# Single-Molecule Fluorescence Probes Interactions between Photoactive Protein—Silver Nanowire Conjugate and Monolayer Graphene

**DOI:** 10.3390/ijms25094873

**Published:** 2024-04-30

**Authors:** Kamil Wiwatowski, Karolina Sulowska, Sebastian Mackowski

**Affiliations:** 1Nanophotonics Group, Institute of Physics, Faculty of Physics, Astronomy and Informatics, Nicolaus Copernicus University, Grudziadzka 5, 87-100 Torun, Poland; kamwiw@gmail.com (K.W.); sulowska@fizyka.umk.pl (K.S.); 2Institute of Advanced Materials, Wroclaw University of Science and Technology, Wybrzeze Wyspianskiego 27, 50-370 Wroclaw, Poland

**Keywords:** fluorescence, energy transfer, plasmonic interaction, graphene, silver nanowire

## Abstract

In this work, we apply single-molecule fluorescence microscopy and spectroscopy to probe plasmon-enhanced fluorescence and Förster resonance energy transfer in a nanoscale assemblies. The structure where the interplay between these two processes was present consists of photoactive proteins conjugated with silver nanowires and deposited on a monolayer graphene. By comparing the results of continuous-wave and time-resolved fluorescence microscopy acquired for this structure with those obtained for the reference samples, where proteins were coupled with either a graphene monolayer or silver nanowires, we find clear indications of the interplay between plasmonic enhancement and the energy transfer to graphene. Namely, fluorescence intensities calculated for the structure, where proteins were coupled to graphene only, are less than for the structure playing the central role in this study, containing both silver nanowires and graphene. Conversely, decay times extracted for the latter are shorter compared to a protein—silver nanowire conjugate, pointing towards emergence of the energy transfer. Overall, the results show that monitoring the optical properties of single emitters in a precisely designed hybrid nanostructure provides an elegant way to probe even complex combination of interactions at the nanoscale.

## 1. Introduction

Among the processes that can induce strong modifications of the optical properties at the nanoscale, two are outstanding: Metal-Enhanced Fluorescence (MEF) and Förster Resonance Energy Transfer (FRET). For each of them to occur, at least two components forming a nanostructure are required, with their mutual distances in the range of tens of nanometers [1,2,3]. Modern fabrication methods allow, however, for obtaining hybrid nanostructures, where both MEF and FRET are simultaneously present. In this way, by combining the proper assembly of components with specific characteristics (absorption, emission, surface functionalization, etc.), multifunctional structures with synergistic effects can be designed. With this approach, the application of hybrid nanostructures in areas like catalysis [4,5], sensorics [6,7], optical limiting [8], energy storage [9], or plasmonics [10,11] has been demonstrated.

The effects of either MEF or FRET, as well as their interplay, are particularly attractive in the context of optically active hybrid nanostructures, where MEF—depending on the distance—can lead to either the enhancement or quenching of fluorescence intensity, while FRET can effectively quench donor fluorescence intensity. At the same time, from the spectroscopic point of view, both processes lead to the shortening of fluorescence decay times [12,13]. The emergence and coexistence of these interactions can be probed using single-molecule fluorescence spectroscopy. Indeed, behaviors observed for individual emitters, when analyzed in a statistically relevant manner, can yield unprecedented details about both the gained functionality and the distributions. The downside of this approach concerns the necessity to work with extremely low concentrations of emitters, which requires efficient fluorescence microscopy methods for detecting weak signals [14]. Nevertheless, as single emitters and their optical properties are sensitive to any interactions in their proximity, they are excellent local probes for interactions at the nanoscale, including MEF and FRET processes [15,16,17,18].

For the MEF effect to occur, metallic nanostructures, characterized with sizes in the range of 1–100 nm, are essential components of a hybrid nanostructure [12]. Metallic nanostructures come in an unaccountable variety of forms and shapes, like nanoparticles (nanospheres, nanocubes, nanodiscs, nanotriangles), island films, nanowires, nanorods, or even patterned nanostructures [19,20,21,22,23,24]. Nanostructures with subwavelength diameters and lengths reaching tens of microns, such as silver nanowires, can also act as waveguides for surface plasmon polaritons suitable for the remote activation and detection of luminescence [25,26,27,28]. 

The MEF process involves interactions between plasmonic excitations in a metallic nanostructure and electronic levels of an emitter placed in its vicinity. It can lead to an increase in fluorescence intensity accompanied by the fluorescence decay time shortening [29]. However, for distances below a few nanometers, the fluorescence intensity may be quenched due to dipole oscillations damping by the nearby metal or by non-radiative energy transfer [2,29]. The fluorescence intensity enhancement occurs mainly due to the increase in the radiative emission rate of the emitter; however, in some cases, the increase in fluorescence intensity may result from plasmon-enhanced absorption [30]. Among the parameters that influence the MEF effect, the distance between the emitter and metallic nanoparticle as well as the matching of the spectra of both components play the key role, and thus, the optimization of these parameters is required for achieving fluorescence enhancement [29,30,31,32,33,34,35]. One approach to control the distance is the conjugation of emitters with chemically functionalized metallic nanoparticles using selective bonding, based on virus-antibody [36,37], streptavidin-biotin [8,38], or other linkers. The fluorescence enhancement has recently been observed for a nanostructure, where streptavidin-terminated photoactive proteins were attached to biotin-functionalized silver nanowires, validating this strategy. Silver nanowires, with their dimensions and relatively large surfaces, in addition with proper functionalization protocols applied, may be rather attractive nanostructures for applications in areas such as sensorics [39,40].

The basic feature of the FRET process is a non-radiative energy transfer between a donor and an acceptor, occurring via dipole–dipole interactions [41]. One of the requirements of FRET is a spectral overlap between acceptor absorption and donor emission. Spectroscopically, the symptoms of FRET are partially contrary to MEF, as it results in the quenching of donor fluorescence intensity as well as the shortening of the fluorescence decay time [42,43,44,45]. The FRET efficiency depends donor–acceptor distance (d), dipole orientation, the spectral overlap integral, and the refractive index of the medium [19,41]. In a classical case, with two interacting dipoles, the efficiency of the energy transfer is proportional to d^−6^ [42], rendering FRET as a “spectroscopic nanoruler”, a tool used to measure distances at the nanoscale [46,47,48,49]. While in a typical configuration, both donor and acceptor are emitting species, sometimes non-emitting materials can be used as energy acceptors. These include dark quenchers [50], carbon nanotubes [51], or graphene [52], which are suitable for FRET architectures as long as their absorption overlaps with the donor emission. 

Taking all these factors into account, graphene, a material composed of carbon atoms arranged in a two-dimensional, hexagonal lattice, is a rather unusual and unique energy acceptor [52,53]. While its extraordinary mechanical [54], thermal [55], and electrical [56] properties are well known, from the point of view of FRET, the optical properties of graphene are essential. Namely, due to the linear dispersion relation near K points of the Brillouin zone [52], the absorption of graphene is constant and equal to 2.3% of incident light across the visible and infrared spectral ranges [57]. Thus, the spectral overlap requirement can be easily fulfilled for many donor molecules. This, together with the absence of graphene fluorescence, renders this material a universal energy acceptor [58,59].

The two-dimensional morphology of a graphene layer influences the FRET distance dependency. While in the case of two interacting dipoles, the efficiency of FRET scales as d^−6^, replacing one of the dipoles with a two-dimensional array of dipoles changes this dependence to d^−4^ [59,60,61,62,63,64,65,66,67]. In addition, it has been shown that the efficiency of the energy transfer is also altered by graphene morphology, showing its decrease with an increasing number of graphene layers [62], as well as the excitation wavelength, since the efficiency is less for longer wavelengths [62]. With graphene as an energy acceptor high energy transfer efficiency can be achieved, exceeding 95% for emitters placed directly on its surface [59,63]. 

The interactions at the nanoscale can be examined and probed through spectroscopic experiments, mainly via measuring fluorescence intensities and decay curves. Is has been shown that MEF and FRET processes may occur for many types of emitters, such as quantum dots [64,68], semiconductor nanocrystals [69,70], organic dyes [60,71], and photoactive proteins [72]. When using graphene as an energy acceptor, the FRET is highly efficient with the almost complete quenching of donor fluorescence [59,63]. However, for photoactive proteins, where emitting molecules, like chlorophylls, are embedded in protein matrix, which protects them from direct contact with graphene [62], the impact of the energy transfer is not so dramatic. In addition, the protein shield inhibits the non-radiative energy transfer present in the MEF process [73]. Consequently, a hybrid nanostructure, where a protein is combined with a plasmonically active metallic nanoparticle and a graphene monolayer, may provide a playground for studying the interplay between these two fundamental nanoscale interactions. 

In this work, we use wide-field and confocal fluorescence microscopy to monitor the influence of both metal-enhanced fluorescence and Förster resonance energy transfer on the optical properties of single photoactive proteins. The sample exhibiting both of these effects consists of silver nanowires, to which proteins were bound using a specific linkage, that were next deposited on a monolayer graphene. Silver nanowires are unique plasmonic nanostructures from the point of view of a very broad plasmon resonance, their lengths that allow for direct observation using an optical microscope, and the ability to functionalize their surfaces. In order to facilitate the comparison and elucidation of the observed effects, appropriate reference samples were also prepared. These include proteins in a polymer layer, proteins deposited on a monolayer graphene, and proteins conjugated to silver nanowires. The results show the fluorescence enhancement of photoactive proteins conjugated with silver nanowires due to plasmonic interactions, as evidenced by the observed shortening of the fluorescence decay time. When photoactive proteins are deposited on graphene, the energy transfer occurs, manifesting itself with fluorescence quenching and decay time shortening. Finally, when both graphene and silver nanowires are assembled together in a hybrid nanostructure, the spectroscopic indications of both MEF and FRET can be observed. The results show that the monitoring the optical properties of single emitters in a precisely designed hybrid nanostructure provides an elegant way to probe even complex combination of interactions at the nanoscale. 

## 2. Results

Representative fluorescence intensity map measured for each studied structure is presented in Figure 1. The images were acquired with the same parameters to facilitate a straightforward comparison. In the case of the first sample, where PCP complexes in PVA are spin-coated on a glass substrate (Figure 1a), we can recognize bright, isolated spots distributed randomly across the substrate. Often, the spatial dimension of these spots is diffraction-limited. They also exhibit some variation in intensity. While some of the brightest spots may originate from aggregates of PCP complexes or locations, where more than a single PCP is present within the area comparable to the spatial resolution, most of the spots seen in the image can be associated with the emission of single-PCP complexes [74]. This is confirmed by movies acquired for this sample, where the clear blinking and photobleaching of PCP fluorescence is observed. The measurement of fluorescence dynamics allows for the unequivocal association of the intensity spots with emission of single PCP complexes, and only such objects were analyzed in the following. The fluorescence intensity map obtained for a sample, where highly diluted PCP complexes were embedded in PVA matrix spin-coated on a monolayer graphene substrate (Figure 1b), is qualitatively similar. For this structure, we also find random distribution of emission spots ascribable to the emission of single-PCP complexes. However, in this case, the average intensity of fluorescence is considerably less than for the PCP complexes on a glass substrate. In accord with the previous studies, where the fluorescence properties of emitters in ensemble concentration were analyzed, this reduction in the fluorescence intensity may suggest the emergence of the energy transfer from PCP complexes to graphene [62], resulting in quenching of emission intensity. 

Importantly, changing the geometry of the hybrid nanostructure by specific conjugation of the PCP complexes with AgNWs yields a qualitative shift in the fluorescence imaging results. Namely, for the sample, where PCP complexes conjugated with silver nanowires were placed on glass (Figure 1c) and on monolayer graphene (Figure 1d), the emission spots are visible only along nanowires. The transmission images corresponding to the fluorescence intensity maps shown in Figure 1 are presented in the Appendix A, and they clearly show the positions of the silver nanowires on the surface. Also, the intensity scale for the images where the fluorescence data obtained for PCP@AgNWs conjugate is displayed is 5-fold expanded. The strong positional selectivity, qualitatively different from the random distribution of emission spots visible in Figure 1a,b, indicates the successful and efficient conjugation of PCP to silver nanowires. Nevertheless, although the emission of PCP complexes is only observed at locations of silver nanowires, the coverage of the nanowires is not uniform. It is expected that the actual coverage depends on the concentration of PCP solution used in the conjugation procedure, which takes place in the solution. When a higher concentration of the photoactive protein is applied, the nanowires are uniformly coated with PCP complexes [8,31].

The initial comparison of the influence of a geometry of a hybrid nanostructure on the optical properties of PCP complexes can be seen using simple cross-sections, as displayed in Figure 2, where we show the fluorescence intensity profiles extracted for each sample. In selecting the profiles, we omitted high-intensity spots that can be due to the presence of PCP agglomerates. Additionally, to make sure the PCP is placed on glass or graphene, we linked each fluorescence intensity map with transmission image, as in Figure 1. All cross-sections shown in Figure 2, regardless of whether they were extracted for PCP on glass (black) or graphene (red), as well as for PCP@AgNWs conjugates on glass (blue) and graphene (green), feature spikes with strong intensity variations from one structure to another. The spikes in emission intensity may be attributed to the emission of single-PCP complexes, as typically, for this PCP concentration, both fluorescence blinking and photobleaching can be observed. Several important observations can be made based on these results. First of all, the fluorescence intensity for PCP placed on graphene (red) exhibits lower values than for the reference sample (black), where proteins were placed on glass substrate. On the other hand, upon conjugation with silver nanowires, the emission intensity of PCP complexes increases substantially, as shown by the blue line in Figure 2b. Last but not least, placing the PCP@AgNWs conjugate on monolayer graphene substrate leads to strong reduction in fluorescence intensity. At this time, however, at least for the cross-sections displayed in Figure 2, the intensities of the PCP@AgNWs conjugate deposited on graphene are comparable to that for single-PCP complexes on a glass substrate (black line in Figure 2a).

Figure 3 shows fluorescence intensity histograms obtained for single-PCP complexes placed on glass (Figure 3a, black) and graphene (Figure 3a, red), as well as for single PCP complexes conjugated with silver nanowires deposited on glass (Figure 3b, blue) and graphene (Figure 3b, green). The fluorescence intensities were extracted for single, well-separated spots in the images by summing the counts over square areas with 8 × 8 pixels in size. A minimum of 50 such spots for each sample were analyzed in order to assure for statistical significance of the result. The distribution of fluorescence intensity of single PCP complexes placed on glass (black) ranges from 40 to 200 kcps with the average value of 100 kcps. The broadening of the distribution may originate from the random orientation of the PCP complexes in the PVA layer. Upon placing onto a monolayer graphene (red), the distribution becomes noticeably narrower, the shape of the distribution becomes Gaussian-like with intensities ranging from 9 to 50 kcps with the average value of 25 kcps. As monolayer graphene has been shown to be an efficient energy acceptor [59,63], the strong reduction in the emission intensity of PCP complexes is a result of the energy transfer. The FRET effect can also explain the narrowing of the intensity distribution, as the key parameter governing the energy transfer efficiency to graphene is the distance, not the orientation of dipole moments of chlorophyll molecules in the PCP. 

In the case of the PCP@AgNWs conjugate (blue), the emission intensities extracted using the same approach as previously used are considerably more than for single PCP complexes deposited on glass. Indeed, their values range from 75 to 300 kcps with the average value of 220 kcps, and the distribution features visible broadening. Placing the PCP@AgNWs conjugate on graphene (green) leads to a reduction in fluorescence intensities, which fall within the 50–250 kcps range with the average value of 150 kcps, with some narrowing of the distribution. The increase in the fluorescence intensity of the PCP complexes upon conjugation with silver nanowires can be interpreted as the result of the MEF process, in accordance with previous results reported for highly concentrated samples [10]. It has been shown that, upon interacting with plasmon excitations in silver nanowires, both the emission and absorption rates of pigments comprising the PCP complex are enhanced, leading to the overall increase in fluorescence intensity. Interestingly, when graphene is introduced into the hybrid structure, in addition to the MEF effect, the FRET process also emerges, which leads to a reduction in fluorescence intensity. While the fluorescence quenching of the PCP conjugated with AgNWs is evident from the histograms shown in Figure 3b, the reduction is less that in the case of pristine PCP complexes. Clearly, for PCP@AgNWs placed on graphene, the competition between the plasmon-induced enhancement (MEF) of emission and reduction in emission intensity due to the FRET process with graphene being the energy acceptor, takes place. In such a scenario, the intensity of emission measured for single-PCP complexes is a net outcome of these two—to some degree competing—effects. 

The results of fluorescence microscopy indicate that, by designing a molecular hybrid nanostructure, we can directly distinguish systems, where either FRET (Figure 3a, red) or MEF (Figure 3b, blue) plays the sole role in determining the optical properties. It is facilitated by including specific components in the nanostructure, namely graphene and silver nanowires in our case, accordingly. Furthermore, the results presented in Figure 3 quantitatively show that the gradual expansion of a hybrid nanostructure with silver nanowires and graphene leads to the interplay between MEF and FRET (Figure 3b, green), which determines the spectroscopic signatures of the PCP proteins at a single molecule level. 

In addition to the changes in the fluorescence intensity of emitters, coupling with plasmonically active nanomaterials may result in reduction in fluorescence decay times due to increase in radiative rate [42]. Similarly, the energy transfer also may cause the shortening of the decay time due to the emergence of new non-radiative recombination channels [42]. Therefore, a spatially and time-resolved fluorescence experiment needs to be carried out for a complete picture of the spectroscopic properties of complex hybrid nanostructures, where the graphene-induced quenching competes with plasmonic enhancement. For that purpose, a TCSPC method adapted to a confocal microscope was used. For every sample, the experiment was performed in the following sequence: first, a fluorescence intensity map was registered to determine the positions of either silver nanowires, graphene flakes, or both on the sample surface. In the next step, fluorescence decays were acquired for tens of emission spots measured for each out of four studied hybrid nanostructures. In Figure 4a and Figure 4b, respectively, we show the typical fluorescence intensity maps of single PCP complexes in a PVA matrix and single PCP complexes conjugated with silver nanowires. As the samples were prepared on substrates, where flakes of monolayer graphene were present, the proper selection of the area of the sample allows for the direct comparison of fluorescence intensity and the decay for PCP complexes coupled and uncoupled to the graphene. The boundaries of graphene in Figure 4a,b are marked with green lines. The fluorescence intensity map for highly diluted PCP complexes in a PVA matrix (Figure 4a) features well-separated spots, some of which—characterized with rather high intensities—are most probably due to PCP aggregates. In contrast, on graphene, the number of high-intensity emission spots is noticeably smaller, and the intensities are also less than for PCP placed on glass. This result is in qualitative agreement with the observations made using wide-field fluorescence microscopy. In the case of PCP complexes conjugated with silver nanowires, as presented in Figure 4b, a major part of the nanowire is located on graphene, with just one end lying on the glass. The difference in the intensity of PCP fluorescence between glass and graphene is, in this case, less noticeable. 

In Figure 4c, we compare the averaged fluorescence intensity decay curves for single-PCP complexes placed on glass (black) and graphene (red) together with single PCP complexes conjugated with AgNWs placed on glass (blue) or graphene (green). Each curve is calculated for a minimum of 30 fluorescence intensity decays (included in Appendix A). While there is some spread of intensities of the decays measured for a given type of nanostructure, the temporal behavior is, in each case, pretty uniform. The fluorescence decay curves for PCP on glass (black) and graphene (red) show an almost a monoexponential character with the latter being shorter, which, together with the quenching of fluorescence intensity by graphene, prove the presence of FRET process from PCP to graphene. For the PCP@AgNWs conjugate on glass (blue), the fluorescence decay is biexponential with the faster component being slightly longer than that single PCP on graphene. We attribute it to the MEF effect of plasmonic excitations in silver nanowires. For the PCP@AgNWs conjugate placed on graphene (green), the faster component of the decay curve is comparable to that for PCP placed on graphene and the curve itself features a biexponential character. As can also be seen, when the conjugate is placed on graphene, a slight shortening of the fluorescence decay occurs. Those features strongly suggest the presence of both MEF and FRET processes and their interplay in graphene-PCP@AgNWs sample. Taken all together, there are obvious spectroscopic indicators of MEF-FRET interplay in proposed hybrid nanostructure composed of photoactive proteins, silver nanowires, and graphene. 

Another facet of the interplay between MEF and FRET in a nanostructure comprising photoactive proteins conjugated to silver nanowires deposited on a monolayer graphene can be visualized using a more quantitative analysis of time-resolved data. In this approach, each fluorescence decay curve was fitted with a biexponential function and the weighted average decay time was calculated. Examples of the fits are depicted in Appendix A. The fluorescence intensity, on the other hand, was obtained by integrating the fluorescence decay curve corresponding to each calculated average decay time. In this way, it is possible to correlate both quantities measured in the same experiment, as shown in Figure 5. The left column in Figure 5 includes the histograms of average calculated fluorescence decay times, while the right column displays the correlation plots of fluorescence intensity and the average fluorescence decay time. The analysis was performed for every sample studied in this work. For single-PCP complexes placed on glass (black), the average fitted decay times are in the range of 1.5–3.5 ns with the average value of 2.2 ns. Such a variation might originate from changes in the local environment of the protein; however, most of the PCP complexes feature decay times close to the average value. Also, as shown in Appendix A, the actual transients are rather uniform across the set of measured decay curves. Upon depositing on graphene, the decays are reduced to the range of 0.2–2 ns, with the average value of 0.7 ns. This shortening is accompanied by a strong reduction in the fluorescence intensity and a narrowing of its distribution. These results further confirm FRET from single PCP complexes to graphene. When PCP complexes conjugated with silver nanowires are placed on glass (blue), the fluorescence average decay time features distribution similar to the graphene sample; however, the average value here is 0.5 ns. Interestingly, in this case, we find fluorescence intensities lower than for the PCP complexes deposited on glass. We attribute this observation as related to the relation between the acquisition time of the TCSPC experiment and the photostability of the PCP complexes on silver nanowires. Namely, in contrast to the wide-field fluorescence imaging, where the data are acquired just over a second, here the acquisition time is 60 s. If, during this time, the molecule photobleaches, then the overall contribution to the total measured fluorescence intensity will be less than in the case, when the molecule emits through the whole acquisition time. Indeed, it has been shown that coupling with metallic nanoparticles, while in the short term, leads to the strong enhancement of fluorescence emission, it frequently suffered from faster photobleaching [32]. Finally, when PCP complexes conjugated with silver nanowires are placed on graphene (green), decay times exhibit the lowest values out of all structures, with an average value of 0.3 ns. Not surprisingly, when taking into account the above comment, the correlation plot also shows the lowest fluorescence intensities, which are around 1000 counts. We assign this effect to the FRET-MEF interplay in hybrid nanostructures composed of single photoactive proteins interacting both with silver nanowires and a graphene monolayer. 

## 3. Materials and Methods

As a model photoactive protein, suitable for both conjugation with silver nanowires and monitoring the interactions in graphene-based hybrid nanostructures, the Peridinin–Chlorophyll–Protein (PCP) photosynthetic complex was selected. It is a simple, small (~4 nm), and water-soluble, light-harvesting complex from *Dinoflagellates*. The X-ray structure of the native PCP, which forms a trimer structure, was resolved with 2 Å resolution [75]. For the experiments described in this work, we used PCP complexes from *Glenodinium*, where the PCP monomers are tagged with streptavidin (BD-Biosciences, San Jose, CA, USA). In this way, a conjugation of PCP with properly functionalized silver nanowires can be facilitated. The PCP monomer contains 2 chlorophyll *a* molecules and 8 peridinins arranged into two almost identical subunits. The protein matrix, which surrounds the pigments, shields them from the environment, and at the same time, provides a scaffold for pigment arrangement. Importantly, the optical properties of PCP make this complex suitable for studying interactions in hybrid nanostructures involving both plasmonically active silver nanowires and graphene. The absorption of PCP is very broad, and by extending from 400 to 650 nm, it overlaps with the plasmon resonance of silver nanowires [8]. The fluorescence emission of PCP occurs at 673 nm. The optical spectra of the PCP in solution are included in Appendix A. This protein has previously been used for exploring the interactions with silver nanowires [76] or graphene [62] at the ensemble level. 

The PCP conjugates with silver nanowires (AgNWs) were prepared using AgNWs synthesized using wet-chemistry approach, as previously described [38]. The typical scanning electron microscopy image of a silver nanowire is shown in Appendix A, while the histograms of the diameters and lengths of silver nanowires are given in Appendix A, respectively. The AgNWs were functionalized with biotin by exchanging PVP polymer with cysteamine. The details of this method were reported previously [38]. For each sample, a proper concentration of PCP (0.02 µg/mL) water solution was prepared. Next, 10 μL of the PCP solution was mixed with 10 μL of AgNWs and incubated for 10 min in order to facilitate bioconjugation. Afterwards, 15 μL of the 0.4% PVA polymer was added to the 15 μL of the PCP@AgNWs conjugate solution. For sample preparation, 30 μL of the PCP@AgNWs conjugate was spin-coated with 17 rps on either glass or graphene substrates.

Monolayer graphene was transferred on glass substrates using a graphene transfer method based on the Polymer-Assisted Transfer approach, as described in [77], with slight modifications. The schematics of this process is shown in Appendix A. The graphene material used for the transfer was a single-layer CVD (Chemical Vapor Deposition) graphene grown on copper foil (Graphene Supermarket, Calverton, NY, USA). It features more than 95% of a monolayer with the remaining part being a multilayer. First a polymer layer, 2% cellulose nitrate from Sigma-Aldrich (Saint Louis, MO, USA), was deposited on graphene. After copper etching, the graphene–polymer stack was washed using deionized water, hydrochloric acid, and ammonia solution. Finally, to dissolve the polymer layer, acetone was used. In the final step, graphene flakes were fished out from the bath using glass substrates. This method, which yields continuous and clean graphene layers, shows high versatility and reproducibility, and does not require specialized equipment. 

Four structures were prepared with highly diluted PCP complexes: (1) proteins in a PVA layer spin-coated on glass; (2) proteins in a PVA layer spin-coated on a graphene monolayer; (3) a protein–silver nanowire conjugate deposited on glass; and (4) a protein–silver nanowire conjugate deposited on a monolayer graphene. The concentration of the PCP protein in a solution was chosen to achieve single emission spots in fluorescence images. The experiments were carried out in an analogous way, as described in [10,62].

The wide-field fluorescence imaging of all structures was carried out using a Nikon Eclipse Ti-U inverted microscope equipped with an oil immersion Plan Apo VC 100x, 1,4 NA objective (Nikon, Tokyo, Japan). For the excitation of fluorescence, a 480 nm LED illuminator was used, the power of which was adjusted to 100 µW. The size of fluorescence intensity maps was 45 × 45 µm. For extracting the emission signal, a combination of a T650LPXR dichroic mirror (Chroma, Bellows Falls, VT, USA) and a set of FEL0650, FELH650 longpass and FB670/10 bandpass filters (ThorLabs, Newton, NJ, USA) placed at the entrance of the detector was used. The signal was detected with an Andor iXonDu-888 EMCCD camera with an amplification factor and an acquisition time of 700 and 1 s, respectively. 

Fluorescence spectra and decay curves were measured with a home-built fluorescence confocal microscope based on Nikon Eclipse Ti-S microscope body. The pulsed diode 485 nm laser (Becker&Hickl, Berlin, Germany) with a repetition rate of 20 MHz and a power of 16 µW was used for excitation. Using a 50/50 beamsplitter, the excitation beam was directed to an oil immersion Plan Apo VC, 60x, NA = 1.4 objective (Nikon, Tokyo, Japan), and focused on a sample to the spot of about 200 nm. The fluorescence signal was collected using the same objective, and after passing through the 50/50 beamsplitter, a confocal aperture, and HQ655LP long-pass filter (Chroma), it was directed to a detector. First, fluorescence intensity maps were measured by correlating the raster movement of piezo-electric stage, on which the sample was placed, with a readout of an SPCM-AQRH-16 avalanche photodiode (PerkinElmer, Waltham, MA, USA). In this case, an additional ET675/20 bandpass filter (Chroma) was used. Fluorescence spectra were detected by Andor iDus DV 420A-BV-CCD camera coupled with an Amici prism. Fluorescence intensity decays were acquired using the Time-Correlated Single Photon Counting (TCSPC) module (SPC-150, Becker&Hickl, Berlin, Germany) with a fast avalanche photodiode as a detector (id Quantique id100-50, Geneva, Switzerland). In these experiments, an additional ET675/20 bandpass filter (Chroma) was used. Such a set of experiments allows for the comprehensive description of metal-enhanced fluorescence, fluorescence resonance energy transfer, and their interplay in a structure composed of silver nanowire–photoactive protein conjugate and graphene. 

## 4. Conclusions

In this work, we probe the interplay between plasmon-enhanced fluorescence and Förster resonance energy transfer with single-molecule fluorescence microscopy and spectroscopy. In order to ensure the presence of both effects in a single hybrid assembly, photoactive proteins were conjugated with silver nanowires and deposited on a monolayer graphene. The results of continuous-wave and time-resolved fluorescence microscopy indicate on the one hand plasmonic enhancement of fluorescence for proteins conjugated to silver nanowires, and—at the same time—energy transfer from the proteins to the monolayer graphene. A comparison with the reference structures leads to the conclusion that monitoring the optical properties of single emitters in a precisely designed hybrid nanostructure provides an elegant way to probe the complexity of interactions at the nanoscale.

## Figures and Tables

**Figure 1 ijms-25-04873-f001:**
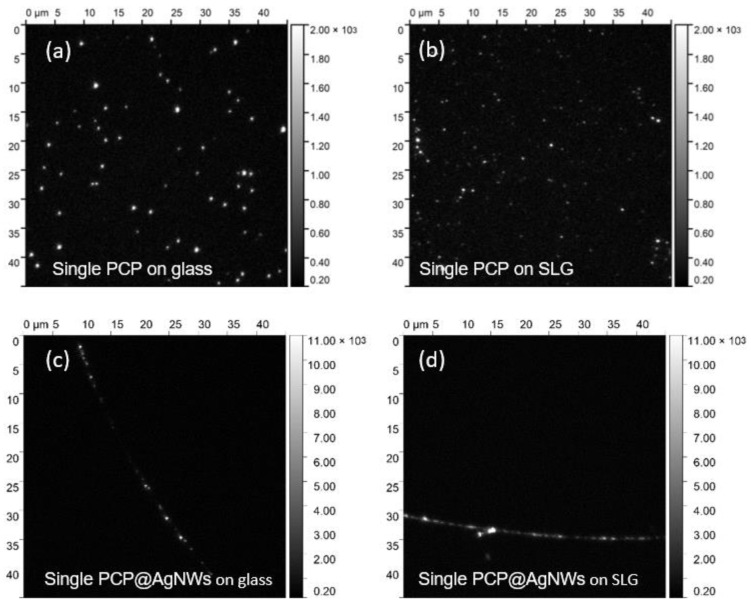
Wide-field fluorescence images measured for single-PCP complexes: (**a**) on glass; (**b**) on graphene; (**c**) conjugated with AgNWs on glass; and (**d**) conjugated with AgNWs on graphene. The corresponding optical transmission images are included in the Appendix A. The intensity scale is different for the data obtained with and without the AgNWs.

**Figure 2 ijms-25-04873-f002:**
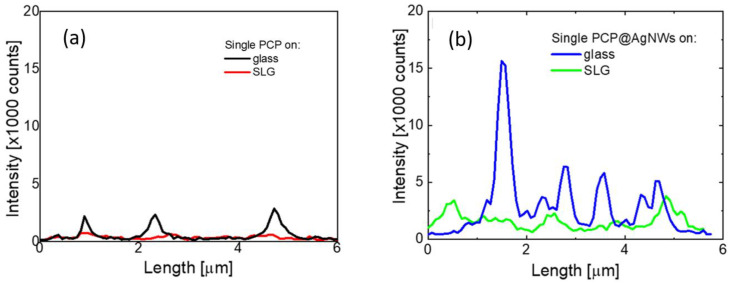
The intensity profiles are as follows: (**a**) single PCP on glass (black) and graphene (red), (**b**) single PCP conjugated with AgNWs on glass (blue) and graphene (green). The profiles for (**b**) were made along AgNWs.

**Figure 3 ijms-25-04873-f003:**
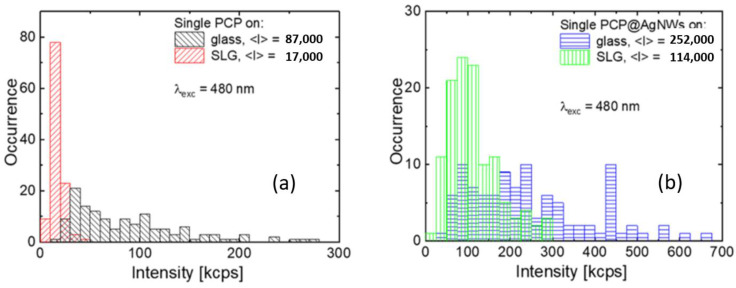
Histograms of fluorescence intensities extracted for the following: (**a**) single PCP on glass (black) and graphene (red), (**b**) single-PCP conjugated with AgNWs on glass (blue) and graphene (green). The average intensity values are included.

**Figure 4 ijms-25-04873-f004:**
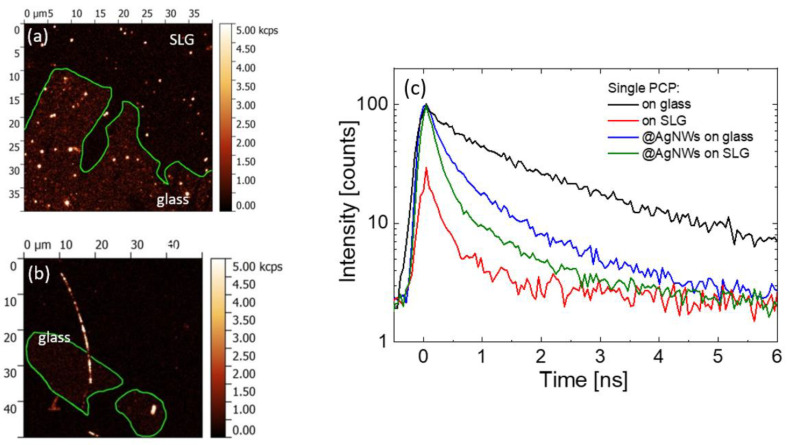
Confocal fluorescence microscopy imaging results are as follows: (**a**) single PCP and (**b**) single PCP conjugated with AgNWs. Graphene–glass boundary was marked with green contour. (**c**) Time-resolved measurements results—fluorescence intensity decays of single PCP: on glass (black), on graphene (red), @AgNWs on glass (blue) and @AgNWs on graphene (green). The presented intensity decays are the averages for all measured data.

**Figure 5 ijms-25-04873-f005:**
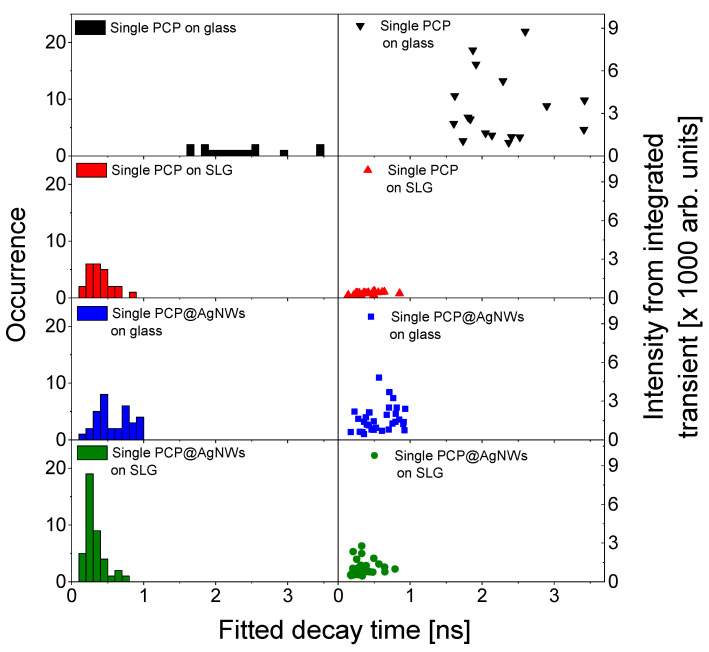
Histograms of fitted average fluorescence decay times (left), integrated fluorescence intensity-average fluorescence decay time correlation plot (right). The data are marked as follows: single PCP on glass (black) and on graphene (red); single PCP@AgNWs on glass (blue) and on graphene (green).

## Data Availability

The original contributions presented in the study are included in the article/Appendix A, further inquiries can be directed to the corresponding author/s.

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
