# Peer review of "Single-Molecule Fluorescence Probes Interactions between Photoactive Protein—Silver Nanowire Conjugate and Monolayer Graphene"

_ijms, 2024, doi:10.3390/ijms25094873_

Round 1
Reviewer 1 Report
Comments and Suggestions for Authors
The manuscript entitled “Single molecule fluorescence probes interactions between photoactive protein – silver nanowire conjugate and monolayer graphene” presents some quite interesting results on applying single molecule fluorescence microscopy and spectroscopy two processes occurring in nanoscale assemblies (namely silver plasmon enhanced fluorescence and Förster resonance energy transfer). The manuscript is well written with conclusions supported by the obtained results. I believe it will be of interest to the readers of the International Journal of Molecular Sciences. However, prior to considering publication there are few points that should be addressed:
1. On figure 1 the authors shown the wide-field fluorescence of PCP complex on 4 different substrates (glass, graphene, AgNWs and AgNWs/graphene). Although clear indication of interaction between the AgNWs and the PCP is observed the coating of the wires are not uniform and even some aggregation is observed (presumably). The author suggested that better coating can be achieved by changing the PCP concentration during the conjugation process. Are there any experiments proving this statement ? If so the authors should provide (may be in supplementary) fluorescence images samples prepared with different concentrations of PCP.
2. On figure 4 and the following text the authors comment of the lifetime measurements and the fact that the AgNWs – containing samples can be fit by biexponential decay fit while the ones on glass and SLG alone by monoexponential. I suggest authors adding some quality data to these statement by including the fitted graphs as well as the R^2. Also the lifetime from different measurements on the same sample shows relatively high variation (for example values between 1.5 and 3.5 ns for PCP on glass). What is the physical meaning behind these differences ?
3. It is unclear to me why exactly the nanowire architecture was chosen and not dots, spheres or any other shape. They can also posses high specific surface area which was one of the stated reason for choosing AgNWs.
4. There are some minor typos in the text that should be addressed e.g. “were” instead of “where”, “pint of view” instead of “point of view” and etc.
In general I find the manuscript to be interesting with some very useful results and it can be considered for publication after addressing the points stated above. Therefore I suggest minor revision.
Author Response
First of all, we would like to thank the Reviewer for positive evaluation of our work. Below we provide detailed response. We believe that upon revision the manuscript is suitable for publication.
The manuscript entitled “Single molecule fluorescence probes interactions between photoactive protein – silver nanowire conjugate and monolayer graphene” presents some quite interesting results on applying single molecule fluorescence microscopy and spectroscopy two processes occurring in nanoscale assemblies (namely silver plasmon enhanced fluorescence and Förster resonance energy transfer). The manuscript is well written with conclusions supported by the obtained results. I believe it will be of interest to the readers of the International Journal of Molecular Sciences. However, prior to considering publication there are few points that should be addressed:
- On figure 1 the authors shown the wide-field fluorescence of PCP complex on 4 different substrates (glass, graphene, AgNWs and AgNWs/graphene). Although clear indication of interaction between the AgNWs and the PCP is observed the coating of the wires are not uniform and even some aggregation is observed (presumably). The author suggested that better coating can be achieved by changing the PCP concentration during the conjugation process. Are there any experiments proving this statement ? If so the authors should provide (may be in supplementary) fluorescence images samples prepared with different concentrations of PCP.
Response: Thank you very much for this comment. The actual fluorescence images obtained for PCP conjugates with silver nanowires where the concentration of the protein is high, are included in the results described in Ref [8] and [31]. In such cases indeed, the nanowires are fully coated with the protein.
- On figure 4 and the following text the authors comment of the lifetime measurements and the fact that the AgNWs – containing samples can be fit by biexponential decay fit while the ones on glass and SLG alone by monoexponential. I suggest authors adding some quality data to these statement by including the fitted graphs as well as the R^2. Also the lifetime from different measurements on the same sample shows relatively high variation (for example values between 1.5 and 3.5 ns for PCP on glass). What is the physical meaning behind these differences ?
Response: Thank you very much for this question. In Fig. S7 we included examples of fits with the quality factors. While the spread of the weighted decay constants for single PCP on glass seems rather broad, it is not completely unexpected. Such a variation might originate from changes in local environment of the protein, as they are embedded in a PVA polymer matrix, and fluorescence decay is more sensitive to any changes in local surroundings that for instance the emission wavelength or its intensity. On the other hand, however, most of the PCP complexes feature decay times close to the average value. Also, as shown in Fig. S6 of Supplementary Material, the actual transients are rather uniform across the set of measured decay curves. We have included text in the revised manuscript to clarify this point, which does not compromise the overall conclusions of our work.
- It is unclear to me why exactly the nanowire architecture was chosen and not dots, spheres or any other shape. They can also posses high specific surface area which was one of the stated reason for choosing AgNWs.
Response: Thank you for this question. This is certainly a valid point. Silver nanowires are unique plasmonic nanostructures from the point of view of a very broad plasmon resonance, which spans from 400 nm to even near-IR spectral region (in fact, plasmons in AgNWrs can be activated using 980 nm). Moreover, their lengths allow for direct observation using an optical microscope, which provides a way to easily locate the nanowires on the substrate. This functionality is utilized in our work. We have included text in the revised manuscript to emphasize this point.
- There are some minor typos in the text that should be addressed e.g. “were” instead of “where”, “pint of view” instead of “point of view” and etc.
Response: Thank you for pointing out these typos. We have read the manuscript carefully and corrected these and couple of other that were found.
In general I find the manuscript to be interesting with some very useful results and it can be considered for publication after addressing the points stated above. Therefore I suggest minor revision.
Reviewer 2 Report
Comments and Suggestions for Authors
This manuscript presents a quantitative analysis of the interaction between Metal-Enhanced Fluorescence (MEF) and Förster Resonance Energy Transfer (FRET) within nanostructures. Specifically, it examines nanostructures where photoactive proteins (PCP) are covalently bound and deposited on monolayer graphene alongside silver nanowires (AgNWs). Prior to publication, it is advisable to address the following minor issues:
1. A schematic diagram illustrating the experimental process would enhance clarity and aid in understanding the methodology.
2. The Introduction section should incorporate the latest literature citation related to protein detection using this method to ensure relevance and completeness.
3. Details regarding the spectral characteristics of the photoactive proteins (PCP) used in the study should be provided to enrich the description of experimental materials.
Comments on the Quality of English LanguageThe quality of English language is good.
Author Response
First of all, we would like to thank the Reviewer for positive evaluation of our work. Below we provide detailed response. We believe that upon revision the manuscript is suitable for publication.
This manuscript presents a quantitative analysis of the interaction between Metal-Enhanced Fluorescence (MEF) and Förster Resonance Energy Transfer (FRET) within nanostructures. Specifically, it examines nanostructures where photoactive proteins (PCP) are covalently bound and deposited on monolayer graphene alongside silver nanowires (AgNWs). Prior to publication, it is advisable to address the following minor issues:
- A schematic diagram illustrating the experimental process would enhance clarity and aid in understanding the methodology.
Response: Thank you for this suggestion. Instead of showing a diagram, we decide to refer to our previous publications, where analogous experiments were carried out. While the current measurements are to some degree different, together with the description provided, we believe that enough details are given regarding the methodology.
- The Introduction section should incorporate the latest literature citation related to protein detection using this method to ensure relevance and completeness.
Response: Thank you very much for this suggestion. There are several articles cited in the manuscript that deal with sensing using various approaches. The ones that are particularly relevant for the discussion presented in our work are [8], [75], [38], and [31].
- Details regarding the spectral characteristics of the photoactive proteins (PCP) used in the study should be provided to enrich the description of experimental materials.
Response: Thank you for addressing this point. We have included appropriate spectra in the Supplementary Material.
Reviewer 3 Report
Comments and Suggestions for Authors
In this study, the authors investigate the possibility of using single-molecule fluorescence microscopy and spectroscopy to study plasmon-enhanced fluorescence and Förster energy transfer in nanoscale assemblies. The authors have found clear evidence of interactions between plasmon enhancement and energy transfer to graphene. Currently, there are several studies in the literature dedicated to investigating this issue with the same objects that the authors used in this study. However, the methods and results presented in this study are original and may be of interest to readers. In its current form, this study requires corrections and cannot be recommended for acceptance.
Major
1. Taking into account the comparability of the light intensity profile of single PCP conjugated with AgNWs on graphene and single PCP on glass, is it reasonable to produce conjugates based on silver and graphene nanofilaments with photochromic proteins for their analysis?
2. There are no absorption and fluorescence spectra of both the PC complex and silver nanowires in the paper. It is required to provide the missing information.
3. There is absolutely no characterization of silver nanowires in the work. It is required to provide this information.
4. The authors should explain the choice of silver nanowires as one of the components of hybrid structures and indicate the advantages of using this particular morphology of nanosilver over others (nanospheres, nanotriangles, nanocubes).
Minor
5. Excessive self-citation by some authors needs to be removed.
6. There is no labeling of the subfigures in Figures 3 , 4.
7. There is no conclusion in the manuscript
8. It is necessary to check the compliance of the list of references with the rules of the journal. Some references are carelessly arranged
9. Abbreviations should be decoded when they are first mentioned (PCP, TCSPC).
Comments on the Quality of English LanguageRequires native speaker text checking
Author Response
First of all, we would like to thank the Reviewer for careful evaluation of our work. Below we respond to all comments and suggestions raised during the evaluation. We believe that upon revision, the manuscript is suitable for publication.
In this study, the authors investigate the possibility of using single-molecule fluorescence microscopy and spectroscopy to study plasmon-enhanced fluorescence and Förster energy transfer in nanoscale assemblies. The authors have found clear evidence of interactions between plasmon enhancement and energy transfer to graphene. Currently, there are several studies in the literature dedicated to investigating this issue with the same objects that the authors used in this study. However, the methods and results presented in this study are original and may be of interest to readers. In its current form, this study requires corrections and cannot be recommended for acceptance.
Major
- Taking into account the comparability of the light intensity profile of single PCP conjugated with AgNWs on graphene and single PCP on glass, is it reasonable to produce conjugates based on silver and graphene nanofilaments with photochromic proteins for their analysis?
Response: Thank you for this question. Actually, assemblies where proteins were conjugated with graphene-based materials have been studied in the past, also in connection with plasmonically active nanostructures. Such structures would however require functionalization of graphene, which is quite an advanced approach, not available in our laboratory. Therefore, we decided to introduce the graphene monolayer in the simplest possible way, that is as a support for protein-nanowire conjugates. Nonetheless, in hybrid nanostructures where proteins were conjugated to graphene, both energy transfer and electron transfer can be observed, which further complicates the interpretation of spectroscopic results.
- There are no absorption and fluorescence spectra of both the PCP complex and silver nanowires in the paper. It is required to provide the missing information.
Response: Thank you for addressing this point. We have included appropriate spectra in the Supplementary Material.
- There is absolutely no characterization of silver nanowires in the work. It is required to provide this information.
Response: Thank you for addressing this point. We have included the results of Scanning Electron Microscopy in the Supplementary Material, including histograms of diameters and lengths of silver nanowires.
- The authors should explain the choice of silver nanowires as one of the components of hybrid structures and indicate the advantages of using this particular morphology of nanosilver over others (nanospheres, nanotriangles, nanocubes).
Response: Thank you for this question. This is certainly a valid point. Silver nanowires are unique plasmonic nanostructures from the point of view of a very broad plasmon resonance, which spans from 400 nm to even near-IR spectral region (in fact, plasmons in AgNWrs can be activated using 980 nm). Moreover, their lengths allow for direct observation using an optical microscope, which provides a way to easily locate the nanowires on the substrate. Also, we have developed protocols for surface functionalization. These functionalities are utilized in our work. We have included text in the revised manuscript to emphasize this point.
Minor
- Excessive self-citation by some authors needs to be removed.
Response: It has been corrected.
- There is no labeling of the subfigures in Figures 3 , 4.
Response: It has been corrected.
- There is no conclusion in the manuscript
Response: We have added conclusions to the revised version of the manuscript.
- It is necessary to check the compliance of the list of references with the rules of the journal. Some references are carelessly arranged
Response: It has been corrected.
- Abbreviations should be decoded when they are first mentioned (PCP, TCSPC).
Response: It has been corrected.
Round 2
Reviewer 3 Report
Comments and Suggestions for Authors
The authors tried to take into account most of the comments and suggestions. The introduction of the manuscript was supplemented with the necessary data justifying the choice of research objects, and typos were corrected. The main characteristics of nano-objects have been added to the SI section. However, a number of points still need to be corrected.
1. Lines 28-37. This text is difficult to read and does not provide any new or specific information. The wording is very streamlined. This fragment needs to be deleted or concretized (what types of structures the authors have in mind, what interactions, etc.).
2. Unfortunately, the authors ignored the comment to remove excessive self-citation of some authors. In particular, the manuscript cited 11 papers of the co-author S. Mackowski, which is 14% of all sources.
Comments on the Quality of English LanguageSome sentences in the introduction are poorly worded. This makes it difficult to understand the text and needs to be corrected.
Author Response
We would like to thank the Reviewer for evaluating our manuscript. Below we respond to the two points that were still outstanding. We believe that in the present form the manuscript is suitable for publication.
- Lines 28-37. This text is difficult to read and does not provide any new or specific information. The wording is very streamlined. This fragment needs to be deleted or concretized (what types of structures the authors have in mind, what interactions, etc.).
Response: this part of the Introduction has been rewritten along the lines suggested. by the Reviewer We believe that right now the wording is streamlined and information is precise.
- Unfortunately, the authors ignored the comment to remove excessive self-citation of some authors. In particular, the manuscript cited 11 papers of the co-author S. Mackowski, which is 14% of all sources.
Response: regarding the number of references, in the first version submitted there were over 20 references that came out from our group. In the revised version we have cut the list to 11, which are essential for the content of the manuscript. While we agree that the amount of references in the first version was far too large, in the case of the present manuscript we are convinced it is just right.
For reference here is the original list:
[2] S. Mackowski, „Hybrid nanostructures for efficient light harvesting”, J. Phys. Condens. Matter, t. 22, nr 19, s. 193102, 2010, doi: 10.1088/0953-8984/22/19/193102.
[3] M. Szalkowski i in., „Plasmonic enhancement of photocurrent generation in a photosystem I-based hybrid electrode”, J. Mater. Chem. C, t. 8, nr 17, s. 5807–5814, 2020, doi: 10.1039/C9TC06983F.
[6] K. Sulowska i in., „Patterned silver island paths as high-contrast optical sensing platforms”, Mater. Sci. Eng. B, t. 268, s. 115124, cze. 2021, doi: 10.1016/j.mseb.2021.115124.
[10] J. Grzelak i in., „Capturing fluorescing viruses with silver nanowires”, Sens. Actuators B Chem., t. 273, s. 689–695, lis. 2018, doi: 10.1016/j.snb.2018.06.119.
[11] M. Szalkowski i in., „Wide-Field Fluorescence Microscopy of Real-Time Bioconjugation Sensing”, Sensors, t. 18, nr 1, Art. nr 1, sty. 2018, doi: 10.3390/s18010290.
[20] S. Mackowski i in., „Energy Transfer in Reconstituted Peridinin-Chlorophyll-Protein Complexes: Ensemble and Single-Molecule Spectroscopy Studies”, Biophys. J., t. 93, nr 9, s. 3249–3258, 2007, doi: 10.1529/biophysj.107.112094.
[22] D. Kowalska i in., „Silver Island Film for Enhancing Light Harvesting in Natural Photosynthetic Proteins”, Int. J. Mol. Sci., t. 21, nr 7, Art. nr 7, sty. 2020, doi: 10.3390/ijms21072451.
[23] D. Kowalska i in., „Metal-Enhanced Fluorescence of Chlorophylls in Light-Harvesting Complexes Coupled to Silver Nanowires”, The Scientific World Journal, mar. 04, 2013. https://www.hindawi.com/journals/tswj/2013/670412/ (udostÄ™pniono sty. 26, 2021).
[27] M. Szalkowski, K. U. Ashraf, H. Lokstein, S. Mackowski, R. J. Cogdell, i D. Kowalska, „Silver island film substrates for ultrasensitive fluorescence detection of (bio)molecules”, Photosynth. Res., t. 127, nr 1, s. 103–108, sty. 2016, doi: 10.1007/s11120-015-0178-x.
[28] N. Czechowski i in., „Absorption Enhancement in Peridinin–Chlorophyll–Protein Light-Harvesting Complexes Coupled to Semicontinuous Silver Film”, Plasmonics, t. 7, nr 1, s. 115–121, mar. 2012, doi: 10.1007/s11468-011-9283-7.
[29] M. Ćwik, D. BuczyÅ„ska, K. Sulowska, E. Roźniecka, S. Mackowski, i J. NiedzióÅ‚ka-Jönsson, „Optical Properties of Submillimeter Silver Nanowires Synthesized Using the Hydrothermal Method”, Materials, t. 12, nr 5, Art. nr 5, sty. 2019, doi: 10.3390/ma12050721.
[30] A. Prymaczek i in., „Remote activation and detection of up-converted luminescence via surface plasmon polaritons propagating in a silver nanowire”, Nanoscale, t. 10, nr 26, s. 12841–12847, 2018, doi: 10.1039/C8NR04517H.
[31] D. BuczyÅ„ska i in., „Correlating Plasmon Polariton Propagation and Fluorescence Enhancement in Single Silver Nanowires”, J. Phys. Chem. C, t. 124, nr 28, s. 15418–15424, lip. 2020, doi: 10.1021/acs.jpcc.0c02364.
[32] S. Mackowski i in., „Metal-Enhanced Fluorescence of Chlorophylls in Single Light-Harvesting Complexes”, Nano Lett., t. 8, nr 2, s. 558–564, 2008, doi: 10.1021/nl072854o.
[34] J. NiedzióÅ‚ka-Jönsson i S. Mackowski, „Plasmonics with Metallic Nanowires”, Materials, t. 12, nr 9, Art. nr 9, sty. 2019, doi: 10.3390/ma12091418.
[41] K. Sulowska, K. Wiwatowski, M. Ćwierzona, J. NiedzióÅ‚ka-Jönsson, i S. Maćkowski, „Real-time fluorescence sensing of single photoactive proteins using silver nanowires”, Methods Appl. Fluoresc., t. 8, nr 4, s. 045004, lip. 2020, doi: 10.1088/2050-6120/aba7cb.
[50] K. Wiwatowski i in., „Energy transfer from natural photosynthetic complexes to single-wall carbon nanotubes”, J. Lumin., t. 170, s. 855–859, luty 2016, doi: 10.1016/j.jlumin.2015.09.034.
[58] S. Mackowski i I. KamiÅ„ska, „Energy Transfer in Graphene-Based Hybrid Photosynthetic Nanostructures”, Recent Adv. Graphene Res., paź. 2016, doi: 10.5772/64300.
[62] I. Kaminska, K. Wiwatowski, i S. Mackowski, „Efficiency of energy transfer decreases with the number of graphene layers”, RSC Adv., t. 6, nr 104, s. 102791–102796, paź. 2016, doi: 10.1039/C6RA20266G.
[63] S. Mackowski i I. KamiÅ„ska, „Dependence of the energy transfer to graphene on the excitation energy”, Appl. Phys. Lett., t. 107, nr 2, s. 023110, lip. 2015, doi: 10.1063/1.4926984.
[70] N. Czechowski, H. Lokstein, D. Kowalska, K. Ashraf, R. J. Cogdell, i S. Mackowski, „Large plasmonic fluorescence enhancement of cyanobacterial photosystem I coupled to silver island films”, Appl. Phys. Lett., t. 105, nr 4, s. 043701, lip. 2014, doi: 10.1063/1.4891856.
[71] M. Kiliszek i in., „Orientation of photosystem I on graphene through cytochrome c553 leads to improvement in photocurrent generation”, J. Mater. Chem. A, 2018, doi: 10.1039/C8TA02420K.
[72] S. Wörmke i in., „Monitoring fluorescence of individual chromophores in peridinin–chlorophyll–protein complex using single molecule spectroscopy”, Biochim. Biophys. Acta BBA - Bioenerg., t. 1767, nr 7, s. 956–964, 2007, doi: 10.1016/j.bbabio.2007.05.004.
[73] M. Twardowska i in., „Fluorescence enhancement of photosynthetic complexes separated from nanoparticles by a reduced graphene oxide layer”, Appl. Phys. Lett., t. 104, nr 9, s. 093103, mar. 2014, doi: 10.1063/1.4867167.
[74] K. Sulowska, K. Wiwatowski, P. Szustakiewicz, J. Grzelak, W. Lewandowski, i S. Mackowski, „Energy Transfer from Photosystem I to Thermally Reduced Graphene Oxide”, Materials, t. 11, nr 9, s. 1567, wrz. 2018, doi: 10.3390/ma11091567.